# Brimonidine and timolol concentrations in the human vitreous and aqueous humors after topical instillation of a 0.1% brimonidine tartrate and 0.5% timolol fixed-combination ophthalmic solution: An interventional study

**Yusuke Orii[1], Eriko Kunikane[2], Yutaka Yamada[1], Masakazu Morioka[1], Kentaro Iwasaki[1], Shogo Arimura[1], Akemi Mizuno[2], Masaru Inatani[1]***

1 Department of Ophthalmology, Faculty of Medical Sciences, University of Fukui, Fukui, Japan, 2 Senju Pharmaceutical Co., Ltd., Osaka, Japan

* inatani@u-fukui.ac.jp

## Abstract

### Purpose

To evaluate the concentrations of brimonidine and timolol in the vitreous and aqueous humors after instillation of a 0.1% brimonidine tartrate and 0.5% timolol fixed-combination ophthalmic solution.

### Methods

This single-arm open-label interventional study included patients with macular holes or idiopathic epiretinal membranes who were scheduled for vitrectomy. Written informed consent was obtained from all participants. A 0.1% brimonidine tartrate and 0.5% timolol fixed-combination ophthalmic solution was administered topically twice daily for 1 week preoperatively. The vitreous and aqueous humors were sampled before vitrectomy, and brimonidine and timolol concentrations were quantified using liquid chromatography-tandem spectrometry. This study was registered with the Japan Registry of Clinical Trials (jRCT, ID jRCTs051200008; date of access and registration: April 28, 2020). The study protocol was approved by the University of Fukui Certified Review Board (CRB) and complied with the tenets of the Declaration of Helsinki.

### Results

Eight eyes of eight patients (7 phakic eyes and 1 pseudophakic eye) were included in this study. The mean brimonidine concentrations in the vitreous and aqueous humors were 5.04 ± 4.08 nM and 324 ± 172 nM, respectively. Five of the eight patients had brimonidine concentrations >2 nM in the vitreous humor, which is necessary to activate α2 receptors. The mean timolol concentrations in the vitreous and aqueous humors were 65.6 ± 56.0 nM and 3,160 ± 1,570 nM, respectively. Brimonidine concentrations showed significant positive

**Data Availability Statement:** All relevant data are within the manuscript and its Supporting Information files.

**Funding:** This study was supported in the form of funding by Senju Pharmaceutical Co., Ltd. (Grant No. University of Fukui J200000772) awarded to MI. The funder had a role in study design and data collection and analysis.

**Competing interests:** The authors have read the journal's policy and have the following competing interests: MI, EK, and AM are paid employees of Senju Pharmaceutical Co., Ltd. This does not alter our adherence to PLOS ONE policies on sharing data and materials. There are no patents, products in development or marketed products associated with this research to declare.

correlations with timolol concentrations in the vitreous humor (P < 0.0001, $R^2$ = 0.97) and aqueous humor (P < 0.0001, $R^2$ = 0.96).

## Conclusions

The majority of patients who received a 0.1% brimonidine tartrate and 0.5% timolol topical fixed-combination ophthalmic solution showed a brimonidine concentration >2 nM in the vitreous humor. Brimonidine and timolol may be distributed in the ocular tissues through an identical pathway after topical instillation.

## Introduction

Glaucoma is an ocular disease that leads to irreversible damage to the optic nerve and loss of vision [1]. There is considerable clinical evidence that lowering intraocular pressure (IOP) is the most effective treatment for attenuating the progression of glaucomatous optic neuropathy [2–4]. While laser or surgical treatments offer IOP reduction, topical application of an ophthalmic solution is the most common initial intervention to lower IOP in patients with chronic glaucoma [5].

Brimonidine is an α2 adrenergic agonist which reduces IOP via two mechanisms: suppression of aqueous humor production and promotion of uveoscleral outflow [6, 7]. Several experimental and clinical studies have suggested that in addition to IOP reduction, brimonidine may provide neuroprotective effects. Furthermore, in patients with open-angle glaucoma treated with prostaglandin-analog ophthalmic solutions, combination therapy with 0.1% brimonidine tartrate (Aiphagan®; Senju Pharmaceutical Co., Ltd., Osaka, Japan) was found reduce progression of visual field loss compared to that with combination therapy with 0.5% timolol [8]. To confirm whether brimonidine was adequately transferred into the vitreous humor to exert a neuroprotective effect on the retinal ganglion cells, several clinical studies have measured the brimonidine concentration in the human vitreous humor after topical instillation [9, 10]. Most eyes administered 0.1%, 0.15, or 0.2% brimonidine tartrate ophthalmic solutions had a concentration of > 2 nM in the vitreous humor, which is known to activate α2 adrenergic receptors in neuronal cells [11].

Currently, most patients with glaucoma in developed countries use multiple IOP-lowering ophthalmic solutions [12]. Multiple topical treatments reduce adherence and increase exposure to preservatives [13, 14]. Compared to separate instillations of agents, medical therapy using a fixed-combination ophthalmic solution is beneficial for avoiding reduced adherence and ocular side effects [15]. Recently, a fixed-combination ophthalmic solution containing 0.1% brimonidine tartrate and 0.5% timolol (equivalent to 0.68% timolol maleate; Aibeta®; Senju Pharmaceutical Co., Ltd., Osaka, Japan) was launched as a new treatment for glaucoma and ocular hypertension in Japan. Regarding brimonidine and timolol fixed-combination therapy, the frequency of allergic conjunctivitis seemed to be lower than that with brimonidine monotherapy [16]. Moreover, the efficiency and absorption of drugs into ocular tissues might be affected by the difference in preservatives and pH values between the fixed-combination and individual drugs, or the interaction between the two components. Therefore, to evaluate the pharmacokinetics of brimonidine and timolol in human eyes, we measured the concentrations in the vitreous and aqueous humors after the instillation of a 0.1% brimonidine tartrate and 0.5% timolol fixed-combination ophthalmic solution.

## Materials and methods

### Patient selection

This single-arm open-label interventional study was approved by the University of Fukui Certified Review Board (CRB) and complied with the tenets of the Declaration of Helsinki. The protocol and possible risks and benefits of the interventions were explained to all the participants before enrollment. Written informed consent was obtained from all participants. This study was registered with the Japan Registry of Clinical Trials (jRCT, ID jRCTs051200008; date of access and registration: April 28, 2020).

Patients scheduled for pars plana vitrectomy to treat macular holes or idiopathic epiretinal membranes from May 2020 to August 2020 were invited to participate in this study. All patients were adults aged ≥20 years. Exclusion criteria were patients with uveitis, vitreous hemorrhage, proliferative diabetic retinopathy, corneal epithelial disorder, a history of allergic reaction to an α2 stimulant or β blocker, or difficulty in instilling ophthalmic solutions.

### Sample collection

The protocol for sample collection was the same as that described in our previous report on brimonidine tartrate (0.1% ophthalmic solution) [17]. Briefly, a fixed-combination ophthalmic solution containing 0.1% brimonidine tartrate and 0.5% timolol (Aibeta®; Senju Pharmaceutical Co., Ltd., Osaka, Japan) was administered topically to the participant, and they were instructed to instill drops in their eyes twice a day for 1 week up to the day before surgery according to the manufacturer's package insert. On the day of surgery, eye drops were administered at 8:00 a.m. and within 2 h before surgery. Patients were instructed to record their adherence to instillation on self-check sheets for 1 week. If adherence was < 75% [18], the patient was excluded from the study.

Vitrectomy was performed under retrobulbar anesthesia using the standard 4-port technique. Vitreous humor (500 μL) and aqueous humor (100 μL) samples were collected from the anterior chamber and vitreous cavity, respectively. To avoid sample dilution, the infusion line was closed until the sample was collected from the vitreous humor. Vitreous humor samples were collected from the vicinity of the retina and optic disc using a 25G vitreous cutter, directed posterior to the optic disc. Samples were stored in Eppendorf tubes at − 80˚C.

### Sample size

In our previous study using 0.1% brimonidine tartrate, subgroup analysis was conducted for five eyes [17]. Also, in the phase 1 trial of Aibeta®, each sample size was between 7 and 9 to evaluate for its safety and pharmacokinetics. Therefore, in this study, the target sample size was 10 patients, including those with withdrawal or protocol breakage during the study. The study was completed when eight samples were collected from patients who completed the protocol.

### Drug concentration measurement

Brimonidine and timolol concentrations in the samples were quantitatively evaluated within 1 month after surgery using liquid chromatography and tandem mass spectrometry in an independent bioanalytical facility (CMIC Pharma Science Co., Ltd., Yamanashi, Japan). A Nexera Ultra High-Performance Liquid Chromatography system (Shimadzu Corporation, Kyoto, Japan) and Triple Quad5500 (AB Sciex Pte. Ltd., Framingham, MA) were used for analysis. Gradient chromatography was performed using an ACQUITY ultra-performance liquid chromatography ethylene-bridged hybrid amide column (2.1 mm I.D. × 50 mm, 1.7 μm; Waters,

Milford, MA), and 5-Chloro-6-(2-imidazolidinylideneamino) quinoxaline was used as an internal standard (IS). The mobile phase consisted of methanol/10 mM ammonium formate (2:3) and acetonitrile, and the flow rate was 0.3 mL/min. Brimonidine, timolol, and IS were analyzed in positive ionization mode with the following multiple reaction monitoring transitions: 292/212 (brimonidine), 317/261 (timolol), and 248/205 (IS). In our previous study [17], we have already checked the drug concentration in vitreous of negative control patients who received no drug instillation and determined values below the lower limits of quantitation.

### Primary outcome measure

The primary outcomes were the mean brimonidine concentrations in the vitreous and aqueous humors and proportion of patients with brimonidine concentration > 2 nM in the vitreous humor, which activates α2 adrenergic receptors of the neuronal cells [11]; and relationship between brimonidine concentrations in the vitreous and aqueous humors.

### Secondary outcome measures

We determined the concentrations of timolol in the vitreous and aqueous humors, relationships between brimonidine and timolol concentrations in the vitreous and the aqueous humors, IOP changes before and after administration of the drug and best-corrected visual acuity (BCVA) before and after surgery. Because patients scheduled for pars plana vitrectomy to treat macular holes or idiopathic epiretinal membranes were not associated with glaucoma, the non-contact tonometer (Nidek, Nagoya, Japan) was used to measure IOPs in order to reduce patient distress. Safety was evaluated throughout the study.

### Statistical analysis

Statistical analyses were performed using the JMP 15 software (SAS Institute, Inc., Cary, NC, USA). Values are shown as the mean ± standard deviation. Correlations between the brimonidine and timolol concentrations in the vitreous and aqueous humors were evaluated using ordinary least-squares regression analysis. IOP change before and after administration was assessed using a paired sample t-test. For all statistical tests, the significance level was set at $P < 0.05$.

Participants, those administering the interventions, and those assessing the outcomes were not blinded.

## Results

### Patients

A total of eight patients were enrolled. No patient discontinued the study protocol (Fig 1). All patients were Japanese, 5 patients were men, and the mean age was 68.3 ± 6.9 years (Table 1). Seven eyes were phakic and one eye was pseudophakic. Seven of the eight patients had an idiopathic epiretinal membrane. The mean adherence rate during the study period was 100%.

### Primary outcome

The mean brimonidine concentrations in the vitreous and aqueous humors were 5.04 ± 4.08 nM (95% CI: 1.62–8.45) and 324 ± 172 nM, respectively (Fig 2). In five of the eight patients (63%), brimonidine concentrations in the vitreous humor were > 2 nM. There was no significant correlation between brimonidine concentrations in the aqueous and vitreous humors (Fig 3A.; $P = 0.93$, $R^2 = 0.001$).

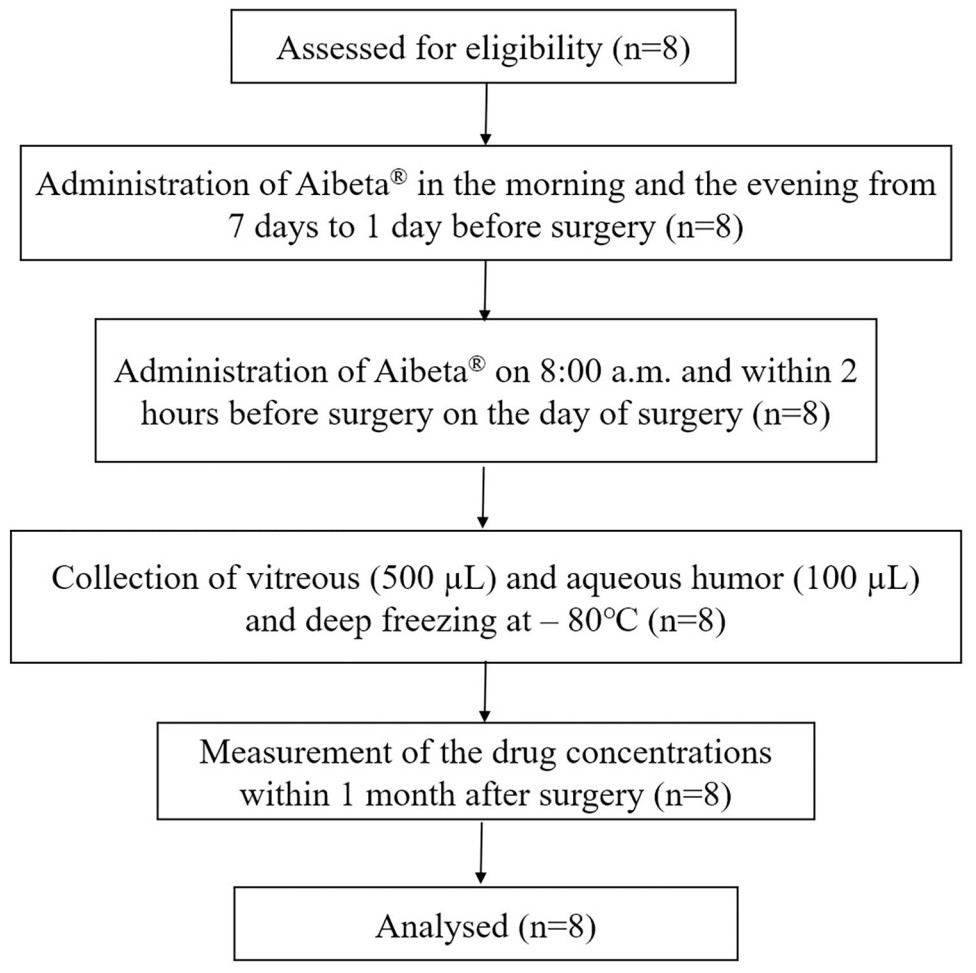

**Fig 1. CONSORT flow diagram.**

## Secondary outcomes

The mean timolol concentrations in the vitreous and aqueous humors were 65.6 ± 56.0 nM and 3,160 ± 1,570 nM, respectively (Fig 4). No significant correlation was found between timolol concentrations in the aqueous and vitreous humors (Fig 3B.; P = 0.92, $R^2$ = 0.002). Significant positive correlations between brimonidine and timolol concentrations were detected in

**Table 1. Patient characteristics.**

| Patients | Values |
|---|---|
| Total (n) | 8 |
| Age, mean ± SD (years) | 68.3 ± 6.9 |
| Sex, male / female (n) | 5 / 3 |
| Lens status, phakia / pseudophakia (n) | 7 / 1 |
| Diagnosis | |
| Idiopathic epiretinal membrane (n) | 7 |
| Macular hole (n) | 1 |

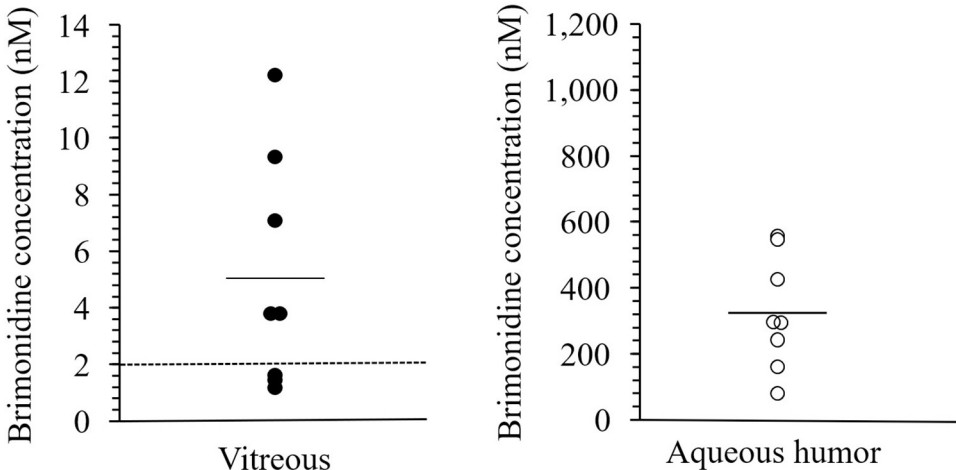

**Fig 2. Brimonidine concentrations in the vitreous and the aqueous humors.** The solid horizontal line in each column indicates the mean of the included data points, and the dotted line parallel to the x-axis denotes the 2 nM brimonidine concentration that was shown to be neuroprotective in animals. The filled and open circles indicate vitreous humor and aqueous humor concentrations, respectively.

the vitreous humor (P < 0.0001, $R^2$ = 0.97) and aqueous humor (Fig 5, P < 0.0001, $R^2$ = 0.96). 95% CI of coefficient of determination were 0.903–0.997 in vitreous and 0.880–0.996 in aqueous humor.

The mean IOP before the start of instillation was 14.3 ±2.9 mm Hg. The mean IOP before surgery was 11.8 ± 2.0 mm Hg, demonstrating a significant difference (p = 0.0094; -2.56 ± 0.72 mmHg; 95% CI: -0.85 to -4.27) in IOP before and after administration. No patient experienced ocular or systemic adverse effects caused by drug administration during the study period. Postoperative complications such as infectious endophthalmitis, vitreous hemorrhage, or retinal detachment were not observed in any patient.

Mean BCVA in logMAR before and 1 month after surgery were 0.31 ± 0.15 and 0.19 ± 0.09. However, there was no significant difference of the BCVA between the two visits (p = 0.0716).

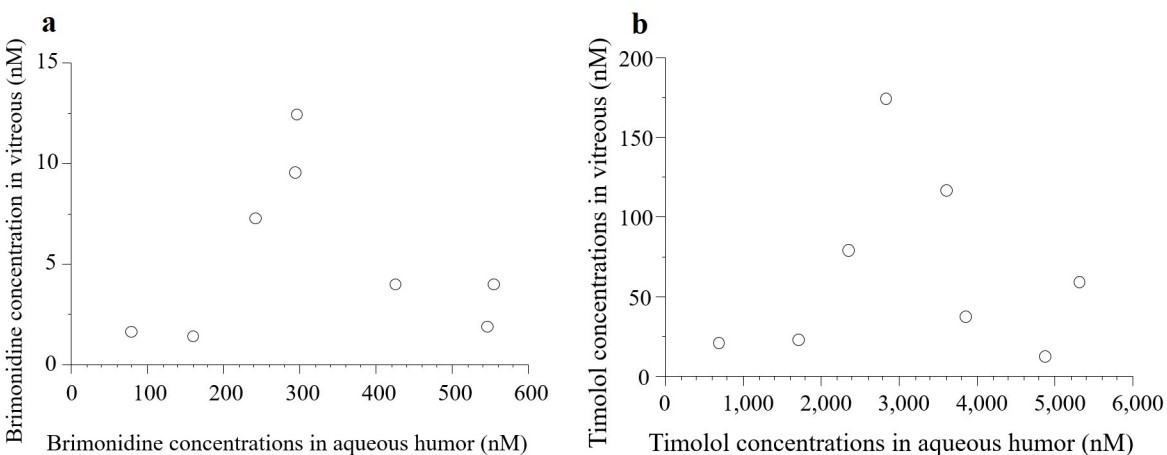

**Fig 3. Correlations between drug concentrations in the aqueous and vitreous humors.** There was no significant correlation of the brimonidine (a) and the timolol (b) concentrations between in the aqueous humor and the vitreous.

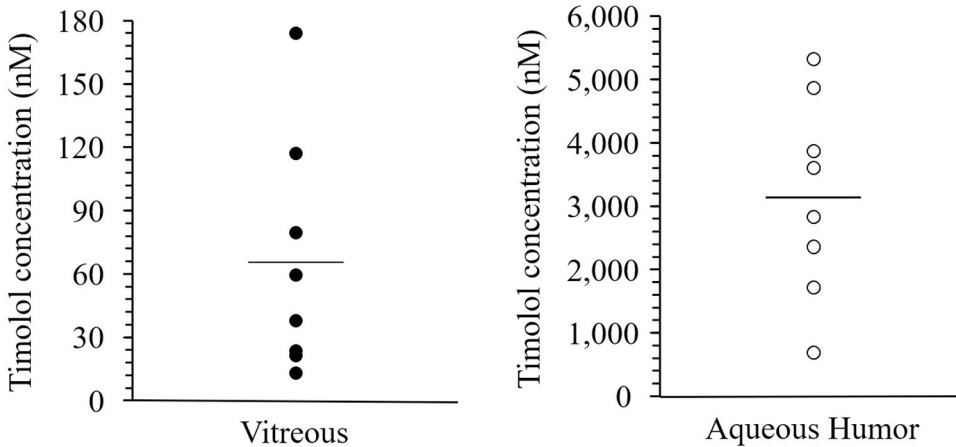

**Fig 4. Timolol concentrations in the vitreous and the aqueous humors.** The solid horizontal line in each column indicates the mean of included data points. The filled and open circles indicate concentrations in the vitreous humor and aqueous humor, respectively.

## Discussion

A 0.1% brimonidine tartrate and 0.5% timolol fixed-combination ophthalmic solution seemed to provide a neuroprotective effect in patients with open-angle glaucoma. To be deemed neuroprotective, an agent must satisfy the following four criteria: (1) receptors on target tissues, such as the optic nerve or retina; (2) adequate penetration into the vitreous humor and retina at pharmacologic levels; (3) induction of intracellular changes that enhance neuronal resistance to insult or interrupt programmed cell death mechanisms in animal models; and (4) demonstration of similar efficacy in clinical trials [19]. Brimonidine met all the criteria [9, 11, 20, 21]. In rats with elevated IOP, subcutaneous injection of brimonidine prevented the loss of retinal ganglion cells without lowering IOP [22]. In a rat model of optic nerve degeneration, brimonidine exhibited neuroprotective effects, whereas timolol did not attenuate degeneration [20]. The Low-Pressure Glaucoma Treatment Study, a randomized clinical trial comparing 0.5% timolol and 0.2% brimonidine tartrate ophthalmic solutions, revealed that in patients with glaucoma, topical treatment with 0.2% brimonidine tartrate (Alphagan®; Allergan, Dublin, Ireland) reduced the frequency of further visual field loss compared to that with 0.5% timolol, despite no significant difference in IOP reduction between the two drugs [21].

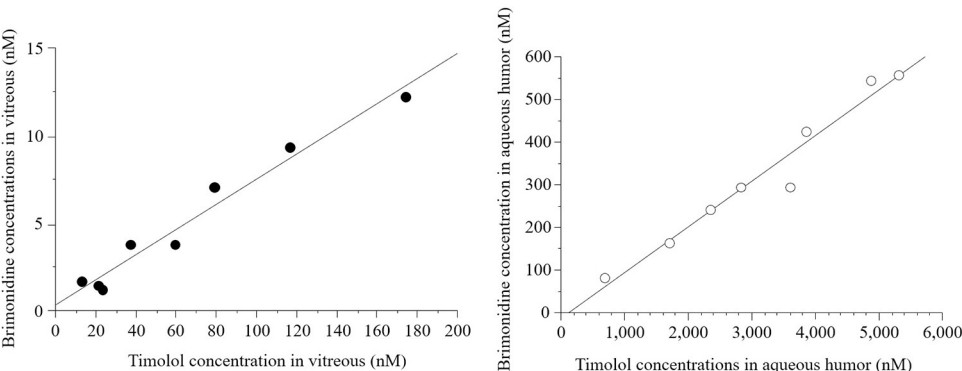

**Fig 5. Linear correlations between brimonidine and timolol concentrations in the vitreous and aqueous humors.** The filled and open circles indicate concentrations in the vitreous humor and aqueous humor, respectively.

The efficacy of fixed-combination therapy is not equal to that of separate treatments. A fixed combination of latanoprost and timolol resulted in a weaker IOP reduction than the concomitant use of the individual components [23, 24]. It is worth noting that the compositions of the 0.1% brimonidine tartrate ophthalmic solution and 0.1% brimonidine tartrate and 0.5% timolol fixed-combination ophthalmic solution are different. The fixed-combination solution contains benzalkonium chloride and ethylendiaminetetraacetic acid, which influence ocular drug pharmacokinetics [25–27], whereas the 0.1% brimonidine tartrate ophthalmic solution does not contain these additives. Further, the pH of 0.1% brimonidine tartrate ophthalmic solution (Aiphagan®) is between 6.7 and 7.5%, whereas the pH of the 0.1% brimonidine tartrate and 0.5% timolol fixed-combination ophthalmic solution (Aibeta®) is between 6.9 and 7.3. The difference in formulation raised concerns regarding the brimonidine concentration in the vitreous humor of eyes treated with the fixed-combination solution. To the best of our knowledge, this study is the first to evaluate brimonidine concentrations in the vitreous humor after topical instillation of a 0.1% brimonidine tartrate and 0.5% timolol fixed-combination ophthalmic solution in human eyes.

Previously, we reported the concentrations of brimonidine in human eyes that received a 0.1% brimonidine tartrate ophthalmic solution twice daily for 1 week, according to the same protocol as that used in the present study [17]. The mean concentrations of brimonidine in the vitreous and aqueous humors of 24 eyes were 4.8 nM and 336 nM, respectively, which are comparable to the brimonidine concentrations after the administration of the fixed-combination solution in the present study (5.04 nM and 324 nM, respectively). The prevalence brimonidine concentrations >2 nM in the vitreous humor was 79% in the previous study and 63% in the present study. The lesser prevalence in the present study seems to be due to the small sample size, because 3 of 8 patients in the present study and 5 of 24 patients in the previous study had a brimonidine concentration < 2 nM in the vitreous humor.

A previous study using a 0.2% brimonidine tartrate ophthalmic solution showed a higher concentration of brimonidine in the vitreous humor of pseudophakic eyes than that in phakic eyes, although the difference was not significant [9]. In this study, we recruited 1 pseudophakic eye. The eye showed brimonidine concentrations of 3.77 nM and 555 nM in the vitreous humor and aqueous humor, respectively. The concentration in the vitreous humor was similar to that in phakic eyes. Our previous study using a 0.1% brimonidine tartrate ophthalmic solution showed no significant difference in brimonidine concentrations in the vitreous humor between phakic (4.9 nM) and pseudophakic eyes (4.1 nM). Independency of lens status might be due to the route of drug penetration into the vitreous humor. To determine the ocular distribution of glaucoma eye drops, radiolabeled nipradiol was topically administered to rabbit eyes. After topical instillation, radioactivity was observed in the anterior chamber, posterior retina, and choroid. However, after intracameral injection of the drug, radioactivity was detected only in the anterior chamber and not in the posterior parts of the eye [28]. These data suggest that drug penetration to the posterior part of the eye is through the periocular tissue, and not across the lens. The hypothesis that brimonidine penetrates through the periocular tissue may be supported by the lack of correlation between brimonidine concentrations in the vitreous and aqueous humors. If brimonidine penetrates from the anterior chamber to the vitreous cavity through the lens, its concentration in the vitreous humor should be correlated with that in the aqueous humor. Although our previous study using a 0.1% brimonidine tartrate ophthalmic solution showed a significant positive correlation between brimonidine concentrations in the vitreous and the aqueous humors, the coefficient of determination ($R^2$) was 0.223. Further studies are required to determine the route of drug penetration into the vitreous humor.

The concentrations of brimonidine and timolol in the vitreous humor and aqueous humor were significantly positively correlated (P < 0.0001, $R^2$ = 0.97; and P < 0.0001, $R^2$ = 0.96,

respectively). These data suggest that the penetration of brimonidine into ocular tissues is similar to that of timolol. An animal study comparing the pharmacokinetics of brimonidine and timolol between the fixed-combination and concomitant instillation of two individual drugs showed similar concentrations in the aqueous humor. However, non-interval concomitant instillation resulted in lower concentrations of brimonidine and timolol in the aqueous humor than that with a 5-min interval between instillation of the two individual drugs [29]. Multiple instillations of eye drops without an appropriate administration interval cause dilution of the drug and reduces its efficiency [30]. The correlation between brimonidine and timolol concentrations in the present study reflects the benefit of the fixed-combination ophthalmic solution, which avoids diluting and washing of each drug.

The present study had some limitations. First, three of the eight patients had brimonidine concentrations <2 nM in the vitreous humor, which was not sufficient to activate α2 receptors in the retina. Assessment of the correlation between patient background and brimonidine concentration in the vitreous humor did not reveal factors affecting the distribution of brimonidine in the vitreous humor. A larger sample size is required to identify these factors. Second, we did not confirm whether the 0.1% brimonidine tartrate and 0.5% timolol fixed-combination ophthalmic solution attenuated the progression of visual field loss in patients with glaucoma, similar to that demonstrated in clinical trials using a brimonidine ophthalmic solution. To identify the neuroprotective effect, a clinical trial comparing a 0.1% brimonidine tartrate and 0.5% timolol fixed-combination ophthalmic solution with another brimonidine-free fixed-combination solution providing comparable IOP reduction would be required in the future. Third, the adherence was self-recorded. Self-recorded adherence is known to be better than actual adherence [31]. In this study, the mean adherence during the instillation period was 100%. It is possible that the actual adherence rate was much lower than 100%.

In conclusion, after twice-daily topical instillation of a 0.1% brimonidine tartrate and 0.5% timolol fixed-combination ophthalmic solution for 7 days, the majority (63%) of patients showed brimonidine concentrations >2 nM in the vitreous humor, which activates α2 adrenergic receptors in neuronal cells. Timolol concentrations in the vitreous and aqueous humors were correlated with brimonidine concentrations, suggesting that the two drugs might share an identical penetration route through ocular tissues.

## Supporting information

**S1 Checklist. CONSORT 2010 checklist of information to include when reporting a randomised trial**\*.
(DOC)

**S1 Data.**
(XLSX)

**S1 File.**
(PDF)

## Author Contributions

**Data curation:** Yutaka Yamada, Masakazu Morioka, Kentaro Iwasaki, Shogo Arimura.

**Writing – original draft:** Yusuke Orii.

**Writing – review & editing:** Eriko Kunikane, Akemi Mizuno, Masaru Inatani.

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
