## [Decision Letter · Decision Letter 0]

21 Aug 2022

PONE-D-22-13271Brimonidine and timolol concentrations in the human vitreous and aqueous humors after topical instillation of a 0.1% brimonidine tartrate and 0.5% timolol fixed-combination ophthalmic solution: An interventional studyPLOS ONE

Dear Dr. Inatani,

Thank you for submitting your manuscript to PLOS ONE. After careful consideration, we feel that it has merit but does not fully meet PLOS ONE’s publication criteria as it currently stands. Therefore, we invite you to submit a revised version of the manuscript that addresses the points raised during the review process.

We look forward to receiving your revised manuscript.

Kind regards,

Callam Davidson

Editorial Office

PLOS ONE

Journal Requirements:

Authors EK and AM are affiliated with Senju Pharmaceutical Co., Ltd. This must be declared as a conflict of interest (via the Submission Form).

Additional Editor Comments (if provided):

PLOS ONE requires that studies be conducted rigorously. Sample sizes must be large enough to produce robust results, where applicable. Experts in the field have reviewed the manuscript and have noted concerns regarding your sample size calculation. Please ensure you address these comments in your rebuttal letter and revised manuscript.

Reviewer #3 suggest the inclusion of a control group, however given that your study was associated with a prospective protocol and inclusion of a control group at this time may prove difficult, an alternative would be to discuss the absence of a control group as a limitation in your Discussion. Any changes to the protocol-specified design or analyses ought to be described as such in the Methods.

Reviewers' comments:

Reviewer's Responses to Questions

**Comments to the Author**

1. Is the manuscript technically sound, and do the data support the conclusions?

Reviewer #1: Yes

Reviewer #2: Partly

Reviewer #3: Partly

2. Has the statistical analysis been performed appropriately and rigorously? 

Reviewer #1: Yes

Reviewer #2: No

Reviewer #3: No

3. Have the authors made all data underlying the findings in their manuscript fully available?

Reviewer #1: Yes

Reviewer #2: No

Reviewer #3: Yes

4. Is the manuscript presented in an intelligible fashion and written in standard English?

Reviewer #1: Yes

Reviewer #2: Yes

Reviewer #3: Yes

5. Review Comments to the Author

Reviewer #1: It is unclear how they calculated sample size for this intervention study. What are the type 1 and 2 error, mean, standard deviation that were used for sample size calculation?

Small sample size of this study is one of the most significant drawbacks that precludes the results convincing.

Furthermore, comparison of drug concentration between phakic and pseudophakic eyes were not performed due to small sample size.

When and how the IOP was measured before and after surgery? Did they consider diurnal variation of IOP? Did they measure IOP with Goldmann applanation tonometry?

Please add best corrected visual acuity and cirumpapillary retinal nerve fiber layer thickness and or macular ganglion cell/inner plexiform layer thickness before and after surgery.

Reviewer #2: The authors reported about a single arm open interventional trial to investigate the concentrations of Brimonidine and timolol in the human vitreous and aqueous humors

after topical instillation. They concluded that the majority of patients who received a 0.1% brimonidine tartrate and 0.5% timolol topical fixed-combination ophthalmic solution showed a brimonidine concentration >2 nM in the vitreous humor.

The present explorative trial needs some minor modifications.

Let me give some detailed comments:

1. The authors are welcome to add individual patient data in a file to facilitate the scientific review process. Note, that the data may be anonymized at this stage.

2. The primary endpoint is not stated. If the primary endpoint variable is "brimonidine concentration >2 nM" the analysis should result in a rate followed by a 95% confidence interval.

3. The presented correlation coefficient should be followed by a 95% confidence interval instead of a statistical test.

4. Please check for decimal point and comma (see line 35).

5. Before after comparison by t-Test (see L141) is difficult to interpret because of regression to the mean and should be avoided. Please use difference of mean with sd and 95% CI for interpretation (see L164).

Reviewer #3: Dear authors,

This manuscript was reviewed properly. This is a small study aiming to determinate the concentration of timolol and brimonidine in the human vitreous body. Although, the design is sound, there are flaws as follows:

1. There are doubts related to sampling:

a. Due to the very spread results of both drug concentrations (mainly in the vitreous body), a control group may help compare very low values.

b. Besides, sample size calculation will also be necessary even considering the explanations made by the authors in the text.

2. A comparison with results from patients treated with non-fixed combination of timolol and brimonidine will be of benefit.

3. Although species and method differences are clear from the Suzuki's and cols. study (reference #29), authors should discuss better the why their study deserves additional publication.

6. PLOS authors have the option to publish the peer review history of their article (what does this mean?). If published, this will include your full peer review and any attached files.

Reviewer #1: No

Reviewer #2: No

Reviewer #3: No

---

## [Author Response · Author response to Decision Letter 0]

29 Sep 2022

Dear Editors and Reviewers.

 I am sending with our fully revised manuscript entitled “Brimonidine and timolol concentrations in the human vitreous and aqueous humors after topical instillation of a 0.1% brimonidine tartrate and 0.5% timolol fixed-combination ophthalmic solution: An interventional study” (Manuscript ID: PONE-D-22-13271), which I would like to submit for publication in PLOS ONE. We have addressed the point raised by referee as much as possible. The comment from reviewer was very valuable and essential to make our manuscript substantially better. We think now that our revised manuscript is worthy to be considered for publication in the great journal, PLOS ONE. Replies for reviewer’s comments are as follows.

Reviewer #1

It is unclear how they calculated sample size for this intervention study. What are the type 1 and 2 error, mean, standard deviation that were used for sample size calculation?

Small sample size of this study is one of the most significant drawbacks that precludes the results convincing.

Furthermore, comparison of drug concentration between phakic and pseudophakic eyes were not performed due to small sample size.

 In the phase 1 trial of Aibeta®, each sample size was between 7 and 9 to evaluate for its safety and pharmacokinetics. We followed the sample size in this study. We added the following sentences in the Materials and Method section, “Also, in the phase 1 trial of Aibeta®, each sample size was between 7 and 9 to evaluate for its safety and pharmacokinetics” (p.7, line 109-110 in the revised version).

As for the comparison of drug concentrations between phakic and pseudophakic eyes, we had examined drug concentrations in phakic eyes as well as pseudophakic eyes ([17] Takamura Y., Tomomatsu T., Matsumura T., “Vitreous and aqueous concentrations of brimonidine following topical application of brimonidine tartrate 0.1% ophthalmic solution in humans.,” J Ocul Pharmacol Ther., 31(5):282-5, 2015.). However, there was no significant difference of the brimonidine concentrations in the vitreous between the 2 groups. We didn’t recruit more numbers of pseudophakic patients to examine whether the vitreous in the pseudophakic eyes contained more concentration of brimonidine. Actually, the concentration of brimonidine in the pseudophakic eye was 3.77 nmol, which was still lower than the mean brimonidine concentration of all the eyes. 

When and how the IOP was measured before and after surgery? Did they consider diurnal variation of IOP? Did they measure IOP with Goldmann applanation tonometry?

Because the main purpose of the present study was to evaluate for the drug concentrations of the vitreous, we did not consider diurnal variation of IOPs to reduce the burden on the recruited patients. Therefore, IOPs were measured without considering the diurnal change in the time-points before the start of the administration as well as before the surgery. We did not use the Goldmann applanation tonometry because of the patients with macular hole or epiretinal membrane. IOPs were measured with non-contact tonometer to avoid patient distress. We described the following sentence in the method. “Because patients scheduled for pars plana vitrectomy to treat macular holes or idiopathic epiretinal membranes were not associated with glaucoma, the non-contact tonometer (Nidek, Nagoya, Japan) was used to measure IOPs in order to reduce patient distress.” (p.9, line 140-142 in the revised version).

Please add best corrected visual acuity and cirumpapillary retinal nerve fiber layer thickness and or macular ganglion cell/inner plexiform layer thickness before and after surgery.

As suggested by the reviewer #1, we added the data for best corrected visual acuity at the following sentences in the revised manuscript. Because of the patients with macular hole or epiretinal membrane, we did not evaluate the cpRNFLT or macular ganglion cell complex to avoid excessive testing. Especially, the thickness of macular ganglion cell complex would not provide reliable data in patients with macular hole or epiretinal membrane. 

In the Secondary outcome measures section: “and best-corrected visual acuity (BCVA) before and after surgery.” (p.9, line139-140 in the revised version)

In the Result section: “Mean BCVA in logMAR before and 1 month after surgery were 0.31 ± 0.15 and 0.19 ± 0.09. However, there was no significant difference of the BCVA between the two visits (p = 0.0716).

” (p.11, line 180 - 181 in the revised version)

Reviewer #2

The authors are welcome to add individual patient data in a file to facilitate the scientific review process. Note, that the data may be anonymized at this stage.

We have added the individual data in a file and upload it. 

The primary endpoint is not stated. If the primary endpoint variable is "brimonidine concentration >2 nM" the analysis should result in a rate followed by a 95% confidence interval.

 As reviewer points out, the primary endpoint is to determine whether the brimonidine concentration in the vitreous is enough (>2 nM).

 We calculated the 95% CI of the brimonidine concentration in vitreous and the result was 1.62-8.45. It spans across 2 nM, but the mean concentration of brimonidine was 5.04 ± 4.08 nM, which is more than 2 nM. This value is almost the same as the result of our previous study with Aiphagan® and majority of patients (63%) showed more than 2 nM of brimonidine in vitreous.

 We added the 95% CI of the brimonidine to the result section: “The mean brimonidine concentrations in the vitreous and aqueous humors were 5.04 ± 4.08 nM (95% CI: 1.62 - 8.45)” (p.10, line 162 - 163 in the revised version)

The presented correlation coefficient should be followed by a 95% confidence interval instead of a statistical test.

 We calculated the 95% CI of coefficient of determination and added the result in revised manuscript: “95% CI of coefficient of determination were 0.903-0.997 in vitreous and 0.880-0.996 in aqueous humor.” (p.10, line 172 - 173 in revised version)

Please check for decimal point and comma (see line 35).

 Thank you for your suggestion. Decimal point and comma are appropriately arranged.

Before after comparison by t-Test (see L141) is difficult to interpret because of regression to the mean and should be avoided. Please use difference of mean with sd and 95% CI for interpretation (see L164).

 We calculated the difference of mean IOP between before and after instillation and 95% CI of the IOP difference. The following sentence is added in the revised manuscript.

 In Results session: “demonstrating a significant difference (p = 0.0094; -2.56 ± 0.72 mmHg; 95% CI: -0.85 to -4.27) in IOP before and after administration.” (p.10, line 175 – 176 in the revised manuscript)

Reviewer #3

1. There are doubts related to sampling:

a. Due to the very spread results of both drug concentrations (mainly in the vitreous body), a control group may help compare very low values.

 In our previous study ([17] Takamura Y., Tomomatsu T., Matsumura T., “Vitreous and aqueous concentrations of brimonidine following topical application of brimonidine tartrate 0.1% ophthalmic solution in humans.,” J Ocul Pharmacol Ther., 31(5):282-5, 2015.), we have already checked the drug concentration in vitreous of control patients who received no drug instillation and determined values below the lower limits of quantitation. To recruit patients during the registration period facing COVID-19 pandemic world-wide, we skipped the negative control group in this present study.

We added the following sentences in the revised manuscript.

In Materials and Methods section: “In our previous study, we have already checked the drug concentration in vitreous of negative control patients who received no drug instillation and determined values below the lower limits of quantitation.” (p.8, line 126-128 in revised manuscript)

b. Besides, sample size calculation will also be necessary even considering the explanations made by the authors in the text.

 As recommended by the reviewer #3, we added the explanations about the sample size in the revised manuscript.

” Also, in the phase 1 trial of Aibeta®, each sample size was between 7 and 9 to evaluate for its safety and pharmacokinetics” (p.7, line 109-110 in the revised version).

2. A comparison with results from patients treated with non-fixed combination of timolol and brimonidine will be of benefit.

As reviewer #3, the data from patients treated with non-fixed combination of timolol and brimonidine would be better. However, we need to minimize the sample size to avoid the risks for adverse effects because the recruited patients were patients with macular hole or epiretinal membrane who had scheduled vitrectomy. Brimonidine frequently causes allergic conjunctivitis and timolol causes bradycardia and worsens asthma and COPD. We did not measure the concentrations of brimonidine and timolol after the instillation of non-fixed combination so that the CRB committee approved the clinical study. 

3. Although species and method differences are clear from the Suzuki's and cols. study (reference #29), authors should discuss better the why their study deserves additional publication.

In animal eyes, the penetrance of the eye drops may be different from human eyes. Therefore, we need to confirm the pharmacokinetics about the brimonidine concentration in the human vitreous. This study is the first to evaluate brimonidine concentrations in the vitreous humor after topical instillation of a 0.1% brimonidine tartrate and 0.5% timolol fixed-combination ophthalmic solution in human eyes. We added the description in the discussion: “To the best of our knowledge, this study is the first to evaluate brimonidine concentrations in the vitreous humor after topical instillation of a 0.1% brimonidine tartrate and 0.5% timolol fixed-combination ophthalmic solution in human eyes.” (p.12, line 210-212 in revised version)

---

## [Decision Letter · Decision Letter 1]

25 Oct 2022

Brimonidine and timolol concentrations in the human vitreous and aqueous humors after topical instillation of a 0.1% brimonidine tartrate and 0.5% timolol fixed-combination ophthalmic solution: An interventional study

PONE-D-22-13271R1

Dear Dr. Inatani,

We’re pleased to inform you that your manuscript has been judged scientifically suitable for publication and will be formally accepted for publication once it meets all outstanding technical requirements.

Kind regards,

Daisuke Nagasato

Academic Editor

PLOS ONE

Additional Editor Comments (optional):

Reviewers' comments:

Reviewer's Responses to Questions

**Comments to the Author**

1. If the authors have adequately addressed your comments raised in a previous round of review and you feel that this manuscript is now acceptable for publication, you may indicate that here to bypass the “Comments to the Author” section, enter your conflict of interest statement in the “Confidential to Editor” section, and submit your "Accept" recommendation.

Reviewer #1: (No Response)

Reviewer #2: All comments have been addressed

Reviewer #4: (No Response)

2. Is the manuscript technically sound, and do the data support the conclusions?

Reviewer #1: Yes

Reviewer #2: Yes

Reviewer #4: Yes

3. Has the statistical analysis been performed appropriately and rigorously? 

Reviewer #1: Yes

Reviewer #2: Yes

Reviewer #4: Yes

4. Have the authors made all data underlying the findings in their manuscript fully available?

Reviewer #1: No

Reviewer #2: No

Reviewer #4: Yes

5. Is the manuscript presented in an intelligible fashion and written in standard English?

Reviewer #1: Yes

Reviewer #2: Yes

Reviewer #4: Yes

6. Review Comments to the Author

Reviewer #1: (No Response)

Reviewer #2: (No Response)

Reviewer #4: (No Response)

7. PLOS authors have the option to publish the peer review history of their article (what does this mean?). If published, this will include your full peer review and any attached files.

Reviewer #1: No

Reviewer #2: No

Reviewer #4: No

---

## [Editor Report · Acceptance letter]

21 Nov 2022

PONE-D-22-13271R1 

Brimonidine and timolol concentrations in the human vitreous and aqueous humors after topical instillation of a 0.1% brimonidine tartrate and 0.5% timolol fixed-combination ophthalmic solution: An interventional study 

Dear Dr. Inatani:

I'm pleased to inform you that your manuscript has been deemed suitable for publication in PLOS ONE. Congratulations! Your manuscript is now with our production department. 

Kind regards, 

on behalf of

Dr. Daisuke Nagasato 

Academic Editor

PLOS ONE